# Risk of dementia in Nepal: A cross-sectional survey in mountainous, hilly, and lowland regions

Bibha Simkhada[1]*, Sanju Thapa Magar[2], Padam Simkhada[3], Pasang Tamang[4], John Stephenson[5]

1 Reader in Nursing and Deputy Director of Graduate Education, Department of Nursing, University of Huddersfield, Huddersfield, United Kingdom, 2 CEO at Ageing Nepal, Kathmandu, Nepal, 3 Associate Dean (International) and Professor of Global Health, School of Human and Health Sciences, University of Huddersfield, Huddersfield, United Kingdom, 4 Lecturer in Public Health, Faculty of Education, Health and Human Sciences, University of Greenwich, London, United Kingdom, 5 Reader in Biomedical Statistics, School of Human and Health Sciences, University of Huddersfield, Huddersfield, United Kingdom

* b.d.simkhada@hud.ac.uk

## Abstract

### Introduction

Globally dementia is a growing public health problem, with over 135,000 people in Nepal living with dementia. Nepal lacks national and community-based data on dementia prevalence. This study aims to determine the dementia risk in Nepal and assess the effects of age, sex, and geographical location on disease prevalence. It also intends to inform policy makers about the burden of dementia, prompting them to plan and prepare appropriate health and social care services for individuals affected by dementia.

### Methods

A cross-sectional survey of 933 older people (aged 60 years and over) was conducted to determine the prevalence of suspected cognitive impairment or dementia risk in three geographical regions of Nepal. The Rowland Universal Dementia Assessment Scale (RUDAS) was used to measure cognitive impairment. The study evaluated the overall prevalence of suspected cognitive impairment or risk of dementia and subgroups by region, age group, and sex. Chi-squared tests and multiple logistic regression analyses were conducted to assess the effects.

### Results

53.7% (501) participants had cognitive impairment or risk of dementia, with slightly higher rates in women (56.5%) than men (51.7%). Cognitive impairment prevalence increases with age and region, with hilly and mountainous areas and low-lying regions having a greater geographic effect.

**Data availability statement:** The datasets used and/or analysed during the current study are available as Supporting information (S1 File).

**Funding:** This study was funded by University of Huddersfield research fund.

**Competing interests:** The authors have declared that no competing interests exist.

## Conclusions

There is high risk of dementia in Nepal. The risk is influenced by age and geographical regions, necessitating early diagnosis and tailored interventions for older people and who are residing in higher altitude areas. Policies implemented to address this issue should prioritise community awareness and early screening and diagnosis to reduce complications from dementia. National-level studies and exploration of factors affecting early dementia diagnosis are needed.

## Introduction

Dementia is a public health problem which requires urgent attention. Globally, the number of elderly people is increasing in proportion to the general population, leading to a corresponding increase in the number of people living with dementia, which is now the 7th leading cause of death worldwide [1]. More than 55 million people around the world are affected by dementia; with about 33 million living in Low- and Middle-Income Countries (LMICs). About 10 million new cases are diagnosed each year worldwide [2]. Correspondingly, the older age population is increasing in Nepal (an LMIC) as elsewhere, where people aged 60 + years old currently comprise about 10% of the total population [3], a 38.2% increase compared to the figure obtained from the previous census of 2011 [4]. It is estimated that over 135,000 people in Nepal may live with dementia, with almost 50% of the population over age of 60 years having some form of memory-related problems [5]. In Nepal, a recent study has determined that nearly 75% of older people living in residential homes in urban areas have dementia symptoms [6] and 93% of older people aged 60 years and above have mild cognitive impairment in one municipality [7]. The situation is compounded in Nepal by a tendency to dismiss symptoms of dementia as part of the normal ageing process, with a lack of awareness of the symptoms of dementia in the general population [8].

Various factors have been reported to be associated with risk of dementia in the literature; for example, gender, age, education, malnutrition, cardiovascular diseases [9–11]. Recent literature also has shown women are disproportionately affected by dementia and the prevalence is higher in women than in men [2,11]. Similarly, age is one of the most cited factors to increase the risk of dementia. The prevalence of dementia increases with age, doubling about every 5 years from 60 years until 85 years of age [11]. Similarly, loneliness is also reported to increase risk of dementia [12]. Recently, there have been massive changes in family dynamics in Nepal, with an increasing number of older people living alone or without their immediate family; circumstances that may have triggered older people to suffer from loneliness, depression and contribute poor health outcomes [13] and to increase the risk of dementia.

There are a few Nepalese studies [14,15] in dementia care and management, but there are no national and community-based data indicating the prevalence of dementia in Nepal [16]. Despite a government policy (Geriatric Health Service Strategy) available for older people's health [17], very limited health facilities are available there. Nearly two-thirds of Nepal's older adults who rely on primary health care

centres (PHCCs) for health services lack basic screening facilities for dementia. A study highlighted the need for community screening procedures for early identification [14]. The Nepalese health system is not fully equipped to deal with or manage the burden of dementia care, and the exact prevalence of dementia is unknown in Nepal. Therefore, this study is the first of its kind in Nepal to assess the prevalence of suspected cognitive impairment or to assess the risk of dementia associated with geographical location (as a proxy for socio-economic and educational capital), gender, and age, which is key to ensure adequate public health policy is in place to support people suffering from this condition.

### Aim

The main aims of this study are to determine the prevalence of suspected cognitive impairment or risk of dementia in Nepal, and assess the effects of age, sex and geographical location (as a proxy for socio-economic and educational capital) on disease prevalence.

The specific objectives of the study are to:

- Compare the prevalences in three geographic regions (hilly, mountainous and low-lying) of Nepal. (regions represent, respectively, regions of medium, low and high socio-economic/educational capital)

- Determine the prevalence of suspected cognitive impairment/ risk of dementia in males and females; and those in different age groups (defined in bands of 5 years from age 60).

- Conduct logistic regression analyses to assess the effect of these factors on dementia/cognitive impairment prevalence.

- Inform policy makers around risk of dementia to provide appropriate health and social care services for people living with dementia.

### Methods

This study employed a cross-sectional research design. Stratified random sampling was utilised to select the study sites by region and the participants for better representation of the subgroups of the population and improved precision of estimates in Nepal [18,19]. Nepal is broadly divided into three ecological regions: the Mountain, Hill, and Terai (Lowland) regions. There are 21 districts in the Mountain, 35 in the Hill and 21 in the Tarai. One district from each of these regions was randomly selected first. The selected districts were Humla from the Mountain region, Kavrepalchowk from the Hilly region, and Siraha from the Lowland (Tarai) region. Following the selection of the districts, a further random selection was made to identify one municipality or rural municipality from each district for the survey. In Humla, which consists of seven rural municipalities, Namkha Rural Municipality was randomly selected. In Kavrepalchowk, out of thirteen municipalities and rural municipalities, Namobuddha Municipality was randomly selected. Similarly, in Siraha, from a total of seventeen municipalities and rural municipalities, Lahan Municipality was randomly selected for the survey. Then a list of households with individuals aged 60 and above was obtained through the relevant municipality to identify eligible participants. The study used the translated Rowland Universal Dementia Assessment Scale (RUDAS) tool to assess cognitive impairment in older adult (60+) from all backgrounds [20]. We used a version of the RUDAS tool translated into Nepalese by Alzheimer's and Related Dementias Society to collect data about performance in everyday living. The RUDAS is a simple tool consisting of six components that assess memory, body orientation, visuospatial praxis, motor praxis, judgment, and language. It has a maximum score of 30, with a recommended cut-off of 22. Scores of 22 or below suggest possible cognitive impairment and referred on for further investigation by the relevant physician. Higher scores indicate better cognitive function, while lower scores suggest poorer ability or risk of dementia. A Computer Assisted Personal Interview (CAPI) was also utilised in the study, which included an option to complete questions from the RUDAS instrument. KoboToolbox, a digital tool for the collection of quantitative data [21], was used as a data collection toolbox.

Although the RUDAS tool is a validated and widely used tool in different cultural settings in many countries and previously translated by Nepal based organisation called 'Alzheimer's and Related Dementias Society', we again tested it at two stages. We initially piloted RUDAS with two older people (60 + years) in Kathmandu using a paper version to understand the examples used in the tools prior to enumerator training. Findings form the pilot study were used to inform changes in the wording of several items to better fit the Nepalese context; without compromising the substantive contribution of each such item to its construct. We also conducted a pre-test of the survey in KoboToolbox within the research team. Following this, we piloted the survey at the study sites with 16 older adults (60 + years). This process helped us confirm their understanding of the survey and provided additional clarifications to the study enumerators on how to input data offline and upload it to the KoboToolbox main survey portal once they have internet access.

All data were collected by trained enumerators between 05/02/2022 and 24/07/2022. Due to the state of cognitive impairment of research participants and the literacy challenges among many individuals, the enumerators completed the survey face-to-face, filling in responses given by participants. Enumerators, who all had prior experience in data collection, received training on dementia and the use of the RUDAS tool to reduce interviewer bias. The training of enumerators was facilitated by the research team which included nursing and public health researchers and a local dementia expert from a local organisation the *Alzheimer's and Related Dementias Society*.

## Statistical analysis

The samples of older people who responded directly (altogether 933 older people, aged 60 and over), were summarised descriptively. The overall prevalence of suspected cognitive impairment was evaluated for the whole sample, and for subgroups defined by region (categorised as low-lying, hilly or mountainous) age group (categorised as 60–64 years, 65–69 years, 70–74 years and 75 + years) and sex. The cognitive impairment was diagnosed by a score of 22 or less in the RUDAS test.

Chi-squared tests were conducted on the data to assess the significance of the age, region and sex factors in uncontrolled analyses. The magnitude of effects observed was quantified using the phi statistic. Multiple logistic regression analyses were conducted to assess the controlled effects of age, sex and region; reporting *p*-values, odds ratios and associated 95% confidence intervals (CIs).

## Ethical considerations

Ethical approval from University of Huddersfield (SREIC/2021/090) and Nepal Health Research Council (Ref: 1303) was obtained prior to the study. All participants were fully informed about the study's purpose, benefits, and potential risks prior to participation, and written consent was obtained from all participants prior to data collection. Older adults (60+) who could understand the study, ask questions, communicate verbally, and provide informed consent were included. All participants were assured that confidentiality, data protection, anonymity would be maintained. They were also explained that they have a right to withdraw from the study without any effect and participation was voluntary.

The enumerators had contact details of the local health facility in case of concern for the health or welfare of participants, or if participants showed the need for psychological support. In such cases, participants were signposted to the nearby health facility for further support.

## Results

### Analysis of data

Data was collected from 933 individuals (549 male) aged 60 years and over from three regions of Nepal: Humla (mountainous region); Kavrepalachowk (hilly region) and Siraha (low-lying region). The sample is summarised in Table 1 below.

Slightly higher prevalences of cognitive impairment were revealed in females than males. Prevalence of cognitive impairment by gender, with associated 95% confidence intervals (CIs), is summarised in Table 2 and Fig 1.

**Table 1. Descriptive summary of sample.**

| Variable | Frequency (valid %) |
|---|---|
| Region | |
| Mountainous | 367 (39.3%) |
| Hilly | 238 (25.5%) |
| Low-lying | 328 (35.2%) |
| Sex | |
| Female | 384 (41.2%) |
| Male | 549 (58.8%) |
| Age group | |
| 60-64 years | 298 (31.9%) |
| 65-69 years | 241 (25.8%) |
| 70-74 years | 204 (21.9%) |
| 75+ years | 190 (20.4%) |

Under the assumption that prevalence of cognitive impairment was diagnosed by a score of 22 or less in the RUDAS test, 501 participants (53.7%) were classified as having cognitive impairment (95% confidence interval 50.5% to 56.9%).

**Table 2. Prevalence of cognitive impairment and associated 95% CIs, by gender.**

| Gender | Prevalence | 95% CI for prevalence |
|---|---|---|
| Males | 284/549; 51.7% | (47.6%, 55.9%) |
| Females | 217/385; 56.5% | (51.6%, 61.5%) |

## Discussion

This study was conducted with 933 individuals aged 60 and older, to determine the prevalence of cognitive impairment or risk of dementia in Nepal. It compared the prevalence of cognitive impairment across different age groups, across male and female gender, and across three geographic regions of Nepal: Humla (mountainous region); Kavrepalachowk (hilly region) and Siraha (low-lying region).

This study revealed that the prevalence of suspected cognitive impairment or risk of dementia in older people was 53.7%; with higher prevalences found in females, in older people, and in those living at higher altitudes (associated with lower levels of socio-economic/educational capital). This prevalence is much lower than recent studies conducted in Nepal, which revealed that 75% of older adults have dementia symptoms using the Cognitive Impairment Test (CIT) tool in residential care homes [6]. Older adults living in residential care homes often have advanced age, multiple health complications and more pronounced cognitive decline, all of which may contribute to the higher prevalence of dementia observed in these settings. Another study conducted in the lowland region of Nepal among 115 older adults (aged 60 and above) found that 93.0% showed signs of mild cognitive impairment using a different tool, the Montreal Cognitive Assessment (MoCA) [7]. The prevalence of cognitive impairment revealed in this study is much higher than our study and a recent systematic review focused on global prevalence, using results from 80 studies which revealed a range of 5% to 41% [22] and in another national study from China, which found a prevalence rate of 22.2% [23]. Therefore, it is interesting to note that the prevalence of cognitive impairment is varied when using different tools and in different settings. Therefore, it is important to understand how this prevalence varied with different tools used and the context. The high prevalence reported in this study could have many explanations, such as low education and social isolation/loneliness is associated with increased risk of dementia [9,12]. The education

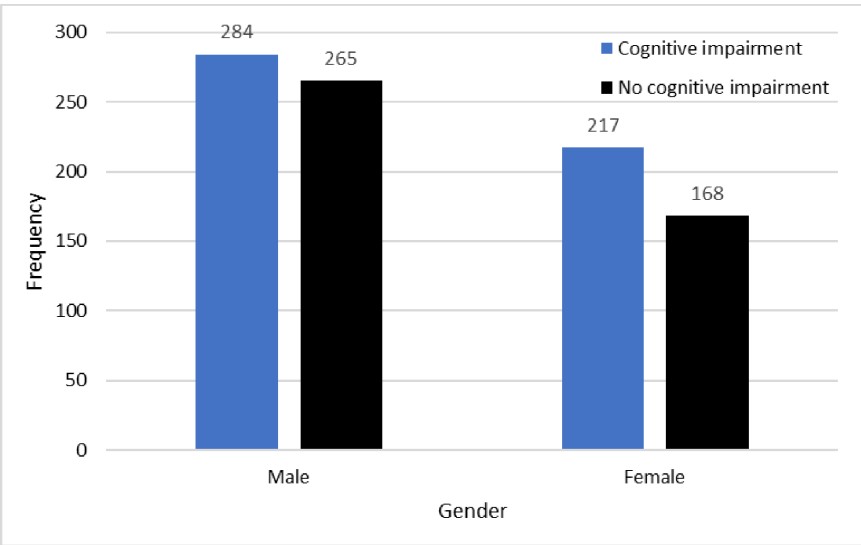

**Fig 1. Cognitive impairment by gender.** The gender effect was not statistically significant at the 5% significance level ($\chi^2_{(1)}$=2.08; $p$=0.150). A strong age effect was observed, with prevalence of cognitive impairment increasing monotonically in age. Prevalence of cognitive impairment by age group, with associated 95% confidence intervals (CIs), is summarised in Table 3 and Fig 2.

**Table 3. Prevalence of cognitive impairment and associated 95% CIs, by age group.**

| Age group | Prevalence | 95% CI for prevalence |
|---|---|---|
| 60-64 years | 131/298; 44.0% | (38.3%, 49.6%) |
| 65-69 years | 110/241; 45.6% | (39.3%, 51.9%) |
| 70-74 years | 126/204; 61.8% | (55.1%, 68.4%) |
| 75+years | 134/190; 70.5% | (64.0%, 77.0%) |

status among older people (60 years and above) in Nepal is very low: almost two thirds of people in this age group are illiterate, and many others can only read and write, without formal education [24,25]. Similarly, due to high levels of internal and international migration [3] resulting massive changes in the family dynamics with an increased number of older people living alone or without their immediate family may have triggered older people to suffer from social isolation/loneliness, depression, and have poor health outcomes [13,26]. The high prevalence of cognitive impairment does not mean that most or all people will develop dementia in later life. However, it may indicate a raised risk for onset of dementia. Therefore, comprehensive diagnoses are required after an indication of depression to rule out other possible problems such as delirium and depression (pseudodementia) in this group. Depression is a risk factor for dementia [27] and very common in older people, ranging from 25% to 60% with depression in the community. This proportion is higher in hospital and residential care homes in Nepal [28,29]. Therefore, the context is important to consider in the diagnosis process.

## Gender

Results from the survey revealed a higher prevalence of cognitive impairment among females than males. Although the difference is not statistically significant, it aligns with previous studies [6,30]. Further, it is important that only few studies have reported significant differences in the prevalence of dementia with respect to gender [31]. As mentioned in earlier

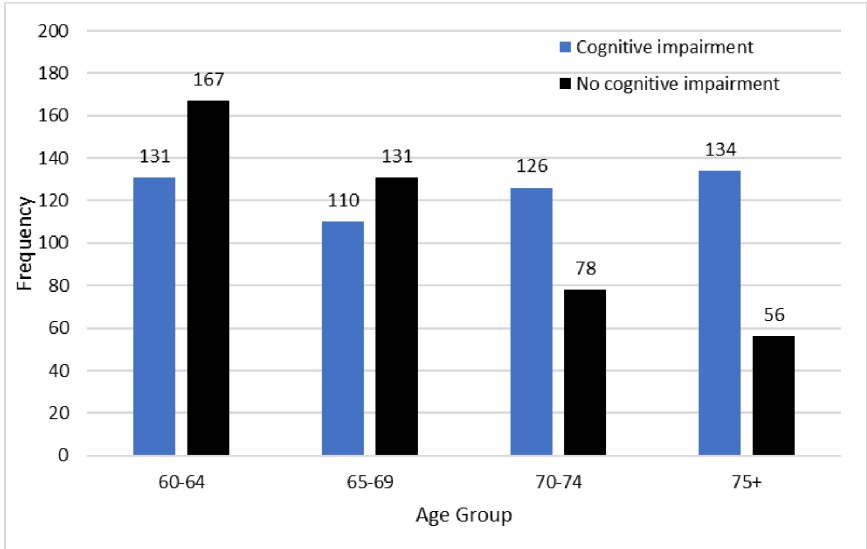

**Fig 2. Cognitive impairment by age group.** The age effect was significant ($\chi^2_{(3)}$=44.6; $p<0.001$) and large in effect ($\varphi=0.219$). A strong geographic effect was also observed, with prevalence of cognitive impairment increasing with higher altitudes. Prevalence of cognitive impairment by region is summarised in Table 4 and Fig 3.

**Table 4. Prevalence of cognitive impairment and associated 95% CIs, by region.**

| Region | Prevalence | 95% CI for prevalence |
|---|---|---|
| Kavrepalanchowk (hilly) | 110/238; 46.2% | (39.9%, 52.6%) |
| Humla (mountainous) | 247/361; 67.3% | (62.5%, 72.1%) |
| Siraha (low-lying) | 75/328; 22.9% | (18.3%, 27.4%) |

section, education may have contributed to the higher prevalence in women, as lower education levels are common among women in this age group due to high illiteracy rates in Nepal, may reduce the brain's capacity to adapt to cognitive decline. Only 48.8% of women in Nepal are literate as compared to 71.7% of males [32]. The World Bank data shows significant disparities in school enrolment between boys and girls in South Asia over the 50 years and earlier [33]. Similarly, women in Nepal live longer [34] and age is a key risk factor for dementia [35]. Gender-related lifestyle factors may impact on dementia to different extents in males and females in certain cultures; such as higher levels of smoking and alcohol consumption reported among males than in females in the Nepalese community [36,37]. Alcohol and smoking are known risk factors for dementia and cognitive impairment [36]. Traditional gender roles, combined with high illiteracy, may have limited women's engagement in cognitively stimulating and social activities, potentially contributing to an increased risk of dementia. There is a need for additional research to understand the impact of gender-related lifestyle factors for cognitive impairment or dementia. Lifestyle-related risk factors can change over time; therefore, a longitudinal cohort study may be useful to understand how gender-related lifestyle factors can impact on the risk of dementia.

## Age

The prevalence of cognitive impairment or risk of dementia increases monotonically with age in this study, reflecting a significant age effect. There is evidence in the literature highlighting the increasing risk of dementia with age [35,38]. Therefore, community-level screening could be one of the best strategies to recognise the risk of dementia as early as

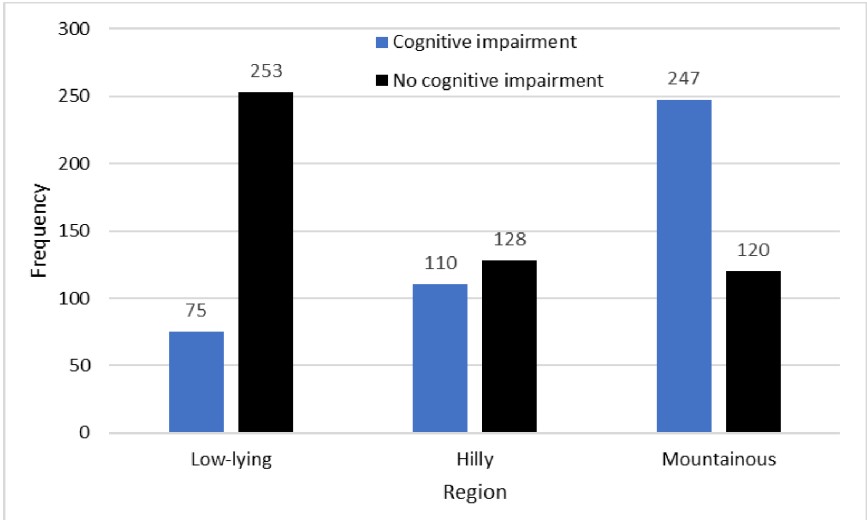

**Fig 3. Cognitive impairment by geographical region.** The geographic effect was significant ($\chi^2_{(2)}$=137.6; $p<0.001$) and very large in effect ($\varphi=0.384$). A multiple logistic regression conducted on the data revealed that in a controlled model, compared to the reference category of age 60-64 years; all age groups were significantly associated with the prevalence of cognitive impairment ($p=0.043$ for age 65-69, $p<0.001$ for other age groups); compared to the reference category of low-lying region, both hilly and mountainous regions were significantly associated with the prevalence of cognitive impairment ($p<0.001$ in all cases); and sex was significantly associated with cognitive impairment ($p<0.001$). Model parameters are summarised in Table 5.

**Table 5. Logistic regression parameters.**

| Variable | *p*-value | Odds ratio | 95% CI for OR |
|---|---|---|---|
| **Sex** | | | |
| Female (reference) | | | |
| Male | <0.001 | 1.83 | (1.34, 2.49) |
| **Age group** | | | |
| 60-64 years (reference) | | | |
| 65-69 years | 0.043 | 1.49 | (1.01, 2.19) |
| 70-74 years | <0.001 | 3.17 | (2.07, 4.83) |
| 75+years | <0.001 | 6.41 | (4.05, 10.2) |
| **Region** | | | |
| Low-lying (reference) | | | |
| Hilly | <0.001 | 3.13 | (2.13, 4.61) |
| Mountainous | <0.001 | 11.8 | (7.97, 17.6) |

possible and it might be beneficial to develop early interventions to reduce the risk. Using new technologies that help to identify biomarkers may be useful in the early diagnosis, prevention, and treatment of cognitive decline [39,40]. The global action plan for a public health response to dementia in the period 2017–2025 also describes goals and suggested courses of action to help nations handle the expanding global dementia crisis [41]. According to the action plan, universal access to care and support for individuals with dementia and their carers depends on prompt and accurate dementia diagnosis, ideally provided at the primary care level. Literature has suggested being cognitively, physically, and socially active from midlife help to reduce the risk of dementia [36]. The awareness programme should focus on risk factors of dementia, recognising early signs and symptoms of dementia and promoting healthy lifestyles [36,41]. Understanding the significance of cardiovascular risk factors that increase the risk of dementia is also important [9,39]; as is the ability to recognise early

signs and symptoms of dementia; and increasing the people's knowledge of risk factors associated with dementia, thereby promoting healthy lifestyles and risk reduction behaviour in all.

## Geographic region

A noticeable geographic effect was observed in this study, with higher prevalence of cognitive impairment associated with living at higher altitudes. This could be associated with rurality, as people living in mountainous area may have many challenges in their daily life due to the local terrain and may also face difficulties in accessing care facilities. The aetiology of dementia may have significant socio-environmental contributions, as suggested by regional variations in dementia prevalence [42]. A systematic literature review comprising 51 studies across the world shown that people living in rural areas had higher prevalence of dementia than those in urban areas [42]. Similar findings were noted among Thai and Chinese people living in rural areas, who had higher prevalence of dementia symptoms [23,43]. The literature suggests that LMICs are not well prepare to care for and support the anticipated future rise in the number of dementia patients and need for advance care planning [44]. Hence, there is a need to prioritise research in rural and underserved areas, as well as low resource settings areas to further establish the relationship between geographic region and dementia, along with the possible explanation for the differences.

## Strengths and limitations

To the best of our knowledge, this is the first study of its kind to compare the prevalence of suspected cognitive impairment or risk of dementia across different geographic locations in Nepal. The findings of this study could help raising awareness about on dementia and prioritise the health and social care demand and preparedness across different geographical regions in Nepal.

This study utilised the most widely used original tool, translated in Nepali to measure cognitive impairment [20] although this tool is translated in Nepali but still to be validated in different context of Nepal. We acknowledge that RUDAS was recently translated and tested in Kathmandu [45] but we recognise that the validation was not entirely appropriate for the wider Nepali context. We observed that item 4 (cube drawing) in Nepali RUSAD was replaced with a matchstick task, we felt cube is better visual-spatial construction, attention, and memory as we were using enumerators. Similarly, in item 5, the traffic light was replaced with a streetlight (*Sadak batti*), that changes the intended meaning. The traffic light is meant to assess judgment for crossing the road safely, whereas a streetlight is primarily for nighttime visibility and safety, not for guiding safe crossing. Therefore, we used a Nepali version translated by professionals working in Demetia care from Alzheimer's and Related Dementias Society. Some adaptations with the Nepali translation were further made only to the example used in the tool without losing meaning of the questions following consultation with a local expert who involved in translation of these Nepali tools to ensure relevancy in the Nepali context: for example, item 5 on judgement in RUDAS elicits a response to the statement *there is no pedestrian crossing and no traffic lights*. This statement is inappropriate to Nepali respondents based outside the capital city of Kathmandu, as there are hardly any roads outside Kathmandu with traffic lights and pedestrian crossings. The item example was accordingly amended. In the same question, typical examples of traffic found in each of the featured geographical areas were included: for example, yaks, sheep, goats, mules in the mountainous area; rickshaws, horse and cart, buses and cars in the hilly area; and buses, trucks, cars and taxis in the low-lying urban area. Similarly, we amended certain items which had been adapted and translated by the Alzheimer's and Related Dementias Society team. It is not considered that any of these amendments have any implications for instrument validity.

This study acknowledges that prevalence estimates may disagree with different scales, as noted in previous studies using various tools. Further research is needed to establish whether the RUDAS self-reporting tool is interpreted correctly in the Nepalese context by participants who record their activity directly using this tool. It is also important to note that there may be variations depending on whether the questionnaire is self-completed by older individuals themselves or completed by trained enumerators.

 

This study has considered the geographical region differences, i.e., mountain, hill and low-lying region. However, there could be other confounding factors such as socio-economic status and health access disparities between regions which have not been considered in this study. Hence, future studies could bring such confounding factors into consideration.

## Conclusion

The prevalence of suspected cognitive impairment or risk of dementia is high in Nepal, with various factors affecting older people's cognitive impairment. The prevalence of cognitive impairment is influenced by age, gender, and geographical location. Therefore, it is important for awareness of dementia and importance of early diagnosis to improve the quality of life of people living with dementia. The significant geographic difference in cognitive impairment calls for tailored interventions and support systems that are specific to geographical location. For example, in mountainous areas, awareness efforts should focus on small, community-led programs, supported by mobile screening services and trained local health workers for early referral and diagnosis. In hill regions, a combination of small and larger group programs including school-based awareness initiatives and strengthened local health services for screening and diagnosis can be effective. In the lowland/Tarai areas, where infrastructure and access are better, large-scale awareness campaigns delivered through schools, regional events, and festival fairs (*melas and mahotsavs*) can be implemented alongside integrated screening and diagnosis by trained healthcare professionals. Across all regions, programs must be culturally appropriate and tailored to local conditions, and should consistently emphasise cognitive health, physical activity, and stigma reduction.

As diverse data is seen in different countries, there is need for national level prevalence studies and exploration of different factors affecting access to early diagnosis of dementia. The recent Geriatric Health Service Strategy indicates Nepal government committed to provide free medical care to older people (17). Therefore, this strategy should be implemented effectively and widely to address this issue and should prioritise community awareness and early screening and diagnosis to reduce complications from dementia.

## Supporting information

**S1 File. SII-RUDAS Data SPSS final Manuscript.**
(SAV)

**S2 File. Inclusivity-in-global-research-questionnaire June 2025.**
(DOCX)

## Author contributions

**Conceptualization:** Bibha Simkhada.

**Data curation:** Sanju Thapa Magar.

**Formal analysis:** John Stephenson.

**Funding acquisition:** Bibha Simkhada.

**Methodology:** Padam Simkhada, Pasang Tamang, John Stephenson.

**Project administration:** Bibha Simkhada.

**Supervision:** Bibha Simkhada, Sanju Thapa Magar, Padam Simkhada, Pasang Tamang.

**Writing – original draft:** Bibha Simkhada.

**Writing – review & editing:** Pasang Tamang, John Stephenson.

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
