## [Decision Letter · Decision Letter 0]

1 Apr 2025

Dear Dr. Simkhada,

Thank you for submitting your manuscript to PLOS ONE. After careful consideration, we feel that it has merit but does not fully meet PLOS ONE’s publication criteria as it currently stands. Therefore, we invite you to submit a revised version of the manuscript that addresses the points raised during the review process.

We look forward to receiving your revised manuscript.

Kind regards,

Saruna Ghimire

Academic Editor

PLOS ONE

2. (1) Please describe in your methods section how capacity to provide consent was determined for the participants in this study. Please also state whether your ethics committee or IRB approved this consent procedure. If you did not assess capacity to consent please briefly outline why this was not necessary in this case.

(2) Please include a complete copy of PLOS’ questionnaire on inclusivity in global research in your revised manuscript. Our policy for research in this area aims to improve transparency in the reporting of research performed outside of researchers’ own country or community. The policy applies to researchers who have travelled to a different country to conduct research, research with Indigenous populations or their lands, and research on cultural artefacts. The questionnaire can also be requested at the journal’s discretion for any other submissions, even if these conditions are not met.  Please find more information on the policy and a link to download a blank copy of the questionnaire here: https://journals.plos.org/plosone/s/best-practices-in-research-reporting. Please upload a completed version of your questionnaire as Supporting Information when you resubmit your manuscript.

 [This study was funded by University of Huddersfield research fund.]. 

4. In the online submission form, you indicated that [The data underlying the results presented in the study are available from lead author.].

Reviewers' comments:

Reviewer's Responses to Questions

**Comments to the Author**

1. Is the manuscript technically sound, and do the data support the conclusions?

Reviewer #1: Partly

Reviewer #2: Yes

Reviewer #3: Partly

2. Has the statistical analysis been performed appropriately and rigorously?

Reviewer #1: Yes

Reviewer #2: Yes

Reviewer #3: Yes

3. Have the authors made all data underlying the findings in their manuscript fully available?

Reviewer #1: Yes

Reviewer #2: Yes

Reviewer #3: Yes

4. Is the manuscript presented in an intelligible fashion and written in standard English?

Reviewer #1: Yes

Reviewer #2: Yes

Reviewer #3: Yes

Reviewer #1: REVIEW:

The paper "Risk of Dementia in Nepal: A Cross-Sectional Survey in Mountainous, Hilly, and Lowland Regions" investigates dementia prevalence among older adults in Nepal, focusing on age, sex, and geographical location. Conducted with 933 participants aged 60 and above, the study utilized the Rowland Universal Dementia Assessment Scale (RUDAS) to measure cognitive impairment. The findings revealed that 53.7% of participants had cognitive impairment or were at risk of dementia, with higher rates in women (56.5%) compared to men (51.7%). The prevalence of cognitive impairment increased with age, and significant geographic variations were observed, with higher prevalence rates in the mountainous (67.3%) and hilly (46.2%) regions compared to the low-lying areas (22.9%).

While the study provides valuable data on dementia prevalence in Nepal, it has several limitations. The validation of the RUDAS tool in the Nepalese context is not thoroughly detailed, raising concerns about the reliability of the findings. The study does not adequately address potential selection bias and unmeasured confounding factors, such as socioeconomic status and healthcare access disparities, which could significantly impact the results. Additionally, the cross-sectional design limits the ability to draw causal inferences about the factors influencing dementia prevalence. I am afraid that selection bias severely disrupts the paper’s ability to draw conclusion on a broader Nepalese population.

The authors emphasize the need for early diagnosis and tailored interventions, particularly for older adults in higher altitude areas. However, the recommendations for public health policy are somewhat generic and lack specific, actionable steps. I am not sure I understand author’s justification of why prevalence is higher in women compared to men in Nepal.

While the paper makes a valuable contribution to understanding dementia risk in Nepal, it requires more rigorous validation of its methods and a deeper exploration of confounding factors to strengthen its conclusions and recommendations.

Reviewer #2: Dear Authors, Great work with the manuscript as it was presented in a coincise manner.

These are my recommendations that i feel will add t the overall quality of the paper and improve comprehension and readability.

Line 74 “preparedness in dementia is deprived” Semantically, Is not fully equipped to deal with or manage the burden of dementia care would be more appropriate.

Line 75 “Estimate” Assess would be a better fit here.

Line 101 “Rowland Universal Dementia Assessment Scale (RUDAS” The reliability and validity measures for the instrument as well as the translated version or any other versions used in literature should be reported as this will show the strength of the instrument used in the paper

Reviewer #3: Thank you for the opportunity to read your work. Overall, this is an interesting and relevant paper aimed to assess dementia risk in Nepal and compare differences based on age, gender and geographic locations.

However, there are a few methodological and minor concerns regarding the methods and results sections. There is an assumption that readers understand the Nepal’s geography, hence very little rationale was presented to support the study aims. It would be helpful to put the study in context for readers to understand the findings. The analysis method is appropriate, although clarification of a few details will be beneficial. The discussion needs an in-depth analysis of the results which will clarify and facilitate understanding of the findings and their implications on dementia risk assessment. Please see the detailed comments attached for your consideration.

**Do you want your identity to be public for this peer review?** For information about this choice, including consent withdrawal, please see our Privacy Policy

Reviewer #1: No

Reviewer #2: **Yes: ** Oluwaseun I Ambode PT, MSc, DPT, PhD(c)

Reviewer #3: No

---

## [Author Response · Author response to Decision Letter 1]

30 Jun 2025

Dear Editor and reviewers,

Thank you for providing us with an opportunity to revise our manuscript. Your insightful comments and suggestions were helpful in improving the quality of our manuscript.

Please find our point-by-point responses to your comments below. These correspond to the track changes in the manuscript, as explained.

Please let us know if any further clarification is needed.

Comments from the Editors and Reviewers:

Editor Comments

Overall comments:

Thank you for the opportunity to read your work. Overall, this is an interesting and relevant paper aimed to assess dementia risk in Nepal and compare differences based on age, gender and geographic locations.

However, there are a few methodological and minor concerns regarding the methods and results sections. There is an assumption that readers understand the Nepal’s geography, hence very little rationale was presented to support the study aims. It would be helpful to put the study in context for readers to understand the findings. The analysis method is appropriate, although clarification of a few details will be beneficial. The discussion needs an in-depth analysis of the results which will clarify and facilitate understanding of the findings and their implications on dementia risk assessment. Please see the detailed comments attached for your consideration.

Introduction

In line 58, Various factors have been reported to be associated with dementia in the literature; Do authors mean the risk of dementia? Please be specific.

Authors: Thank you for the suggestion. We have now corrected to “risk of dementia”

In line 76: authors provide evidence throughout the introduction to support risk factors including age and gender for dementia risk assessment. However, it is unclear why geographical location is considered a risk factor to be assessed in this study. Please consider elaborating this a bit more for clarity. For international readers, it would be beneficial to provide a context of the Nepalese regions and a rationale for why the study has to be conducted in the regions with Hilly, mountainous and low lying areas.

Authors: We have clarified in the text that geographical location is used as a proxy for socio-economic and education.

Aim: There is an assumption that geographic location may be a contributory factor to the risk of dementia. However, this is not clearly stated or supported in the introduction. Thus, comparing the prevalence in three geographic regions is unclear.

Authors: We have clarified in the text that geographical location is used as a proxy for socio-economic and education.

I would be cautious to use the word prevalence if the diagnosis of dementia or cognitive impairment is not confirmed. Perhaps dementia risk will be most appropriate.

Authors: We agree with the suggestion and “prevalence of cognitive impairment” have been revised to “prevalence of suspected cognitive impairment,” and we kept the term “risk of dementia” as it refers to potential risk and not a formal diagnosis. The text has been amended to clarify where a formal diagnosis is not implied.

In line 85: consider reframing the objective without the statistical analysis.

Authors: This is an objective, rather than an aim, so we consider that the text as it stands is appropriate and offer no amendments.

Design & Methods

There is an assumption that readers know regions in Nepal. How many regions are in Nepal? How many of these regions are in the Hilly, mountainous and low-lying regions? Without the context, it is challenging to comprehend how the stratified sampling was done.

Authors: Thank you for your valuable feedback. We acknowledge the need to provide clearer geographical context. We have now added “Nepal is broadly divided into three ecological regions: the Mountain, Hill, and Terai (Lowland) regions. There 21 districts in the Mountain, 35 in the Hill and 21 in the Tarai” in the revised manuscript to provide the context of the region.

Furthermore, the rationale and methodology of our sampling strategy are detailed as described - “ Then a list of households with individuals aged 60 and above was identified through relevant municipality authorities to reach eligible participants. Older adults (60+) who could understand the study, ask questions, communicate verbally, and provide informed consent were included”.

How was the random selection of districts done. How many districts are there in a region? Did authors use a sampling frame for the stratification? If so, how was the frame developed?

Authors: Information already provided: Refer to lines 95-105

My understanding from the methods section is that this was a three-stage stratified sampling process. First, regional sites, district and municipality selections. However, it is not clear how older adults 60+ years were selected. Please make clear how the participants were recruited for the study. What was the inclusion criteria for older adults participating in the study? Did older adults with mild and moderate cognitive impairment participate in the study?

Authors: We added the following sentence: “Once a list of households with individuals aged 60 and over was identified, older adults aged 60+ who were able to understand the study, ask questions, communicate verbally, and provide informed consent were included” is added to clarify the sample inclusion criteria in second paragraph of method section.

Survey instrument: Please elaborate a bit about the Rowland Universal Dementia Assessment Scale (RUDAS) tool. What items are used to measure prevalence of cognitive impairment? Is this a Likert or a dichotomous scale. Is the tool unidimensional or does it have domains? How is the tool rated and summed? How are the scores categorized eg mild, moderate or severe cognitive impairment. What is the reported reliability of the tool? How was the Kobo Toolbox used for data collection. Please clarify to facilitate understanding.

Was the data collected in a different language than English?

Authors: Information already provided: refer to lines 108-109 in previously submitted version.

What informed the sample size of 933? Or how did authors attain a sample size of 933?

Authors: We calculated a sample size based on a two-sided 95% confidence interval with a half-width equal to 5%, based on the main outcome of cognitive impairment, and conservatively basing the calculation on an anticipated prevalence of 50%. This yielded a sample size requirement of 402. To account for 20% attrition, we aimed to approach a minimum of 503 individuals, following the method outlined above. Our actual sample size of 933 meets this minimum requirement.

Line 126-130 should be moved to the methods and not statistical analysis sections.

Authors: We appreciate the reviewer’s observation. However, we respectfully disagree: the text refers to statistical methods and is correctly placed in a subset of the methods section as it stands.

Consider elaborating a bit more about how the statistical analyses were done. Eg How many variables were entered for the logistic analysis? What variable(s) was the outcome variable?

Authors: We appreciate the reviewer’s suggestion. However, we believe this information is already clearly provided in the statistical methods section. We offer no amendments in response to this comment.

Ethical approval

In line 142: Please check spelling “ensured”

Authors: Thank you for highlighting the typo which has been changed to “assured”.

Results

While the findings are interesting, the results are a bit challenging to comprehend given the limited description of the survey tools.

Authors: The following sentences have been added to the Methods section to provide a detailed description of the survey tool “The RUDAS is a simple tool consisting of six components that assess memory, body orientation, visuospatial praxis, motor praxis, judgment, and language. It has a maximum score of 30, with a recommended cut-off of 22. Scores of 22 or below suggest possible cognitive impairment and referred on for further investigation by the relevant physician. Higher scores indicate better cognitive function, while lower scores suggest poorer ability or risk of dementia”.

Discussions

It would be interesting to see the specific variations in geographic location on dementia risk in Nepal.

Authors: This information is already provided in some detail in the results section and additional summary text has now been provided in the discussion section.

In line 190: the 75% prevalence of dementia symptoms were with older adult residents in LTC. Your study was conducted with community dwelling older adults. Please elaborate a bit about this study outcome. Consider other factors that could account for the high prevalence of dementia symptoms in residential homes compared to your finding.

Authors: Thank you for this suggestion. We have now added “Older adults living in residential care homes often have advanced age, multiple health complications, and more pronounced cognitive decline, all of which may contribute to the higher prevalence of dementia observed in these settings” to explain our prevalence.

In line 193: what was the result of the study and what does the finding mean for your result?

Authors: We have discussed the findings of the cited study in relation to our results as outlined - “Another study conducted in the lowland region of Nepal among 115 older adults (aged 60 and above) found that 93.0% showed signs of mild cognitive impairment using a different tool, the Montreal Cognitive Assessment (MoCA)”.

Strength & limitations: This is confusing given that item description was inadequately explained in the methods section. If an elaborate description of the scale was presented earlier, it would have made sense why the scale adaptation would be a strength or limitation.

Authors: Thank you for your suggestion. We have now included the details of the survey tool in the Methods section. We hope this clarification improves the understanding of our study.

In line 250: I was curious to see what could account for the geographic variations in dementia risk especially for older adults in Nepal. However, supporting your findings with rurality in general, maybe inadequate to highlight the specific dementia risk for older adults in these geographic locations in Nepal. If there are no supporting evidence for dementia risk in specific locations, like hilly, mountainous and low-lying areas, it would be worthwhile to reframe the topic and objectives in line with rurality.

Authors: See responses to previous comments

Conclusion

While there were significant dementia risk variations in geographic locations, I wondered what specific tailored care could be developed for older adults in mountainous regions compared with those in low lying regions. Authors: We have now added a few sentences to explain the need to tailor the program specifically for older adults from different geographic regions.

Reviewer 1 comments

The paper "Risk of Dementia in Nepal: A Cross-Sectional Survey in Mountainous, Hilly, and Lowland Regions" investigates dementia prevalence among older adults in Nepal, focusing on age, sex, and geographical location. Conducted with 933 participants aged 60 and above, the study utilized the Rowland Universal Dementia Assessment Scale (RUDAS) to measure cognitive impairment. The findings revealed that 53.7% of participants had cognitive impairment or were at risk of dementia, with higher rates in women (56.5%) compared to men (51.7%). The prevalence of cognitive impairment increased with age, and significant geographic variations were observed, with higher prevalence rates in the mountainous (67.3%) and hilly (46.2%) regions compared to the low-lying areas (22.9%).

While the study provides valuable data on dementia prevalence in Nepal, it has several limitations. The validation of the RUDAS tool in the Nepalese context is not thoroughly detailed, raising concerns about the reliability of the findings.

Authors: We used a widely validated tool that has been applied in over 25 countries. However, for our study, we used a translated version provided by a community organization supporting people living with dementia in Nepal. The rationale for choosing this version is explained on Strengths and Limitations section of the discussion. We also recommend further validation of this translation for broader use in Nepal, particularly in relation to ensuring judgment accuracy and safety within the existing infrastructure.

The study does not adequately address potential selection bias and unmeasured confounding factors, such as socioeconomic status and healthcare access disparities, which could significantly impact the results.

Authors: The geographical location factor is a proxy for known disparities in socio-economic and educational capital in Nepal. The reviewer will be aware that all prevalence studies potentially may be at risk of confounding bias; and no study has the resources, nor even the ability, to identify and quantify all known and unknown confounders. The reviewer will also be aware that random allocation is the only way to account for unknown or unmeasured confounding factors, so a criticism of any non-randomised study for failing to do this is absurd: it amounts to a statement that no clinical or Public Health study except RCTs is worth doing. Criticism of a study for failing to account for unknown confounders suggests significant misunderstanding of the purposes of a population survey. Further, we focussed on variables with which a strong theoretical and evidence base exists for the associations of interest. We are not aware of any readily identifiable, or measurable, potential confounding factors that could have been added to the data collection process.

Additionally, the cross-sectional design limits the ability to draw causal inferences about the factors influencing dementia prevalence.

Authors: We do not draw causal inferences; we state associations. Therefore, we believe no changes are necessary in response to this point.

I am afraid that selection bias severely disrupts the paper’s ability to draw conclusion on a broader Nepalese population.

Authors: We used stratified random sampling to select geographical locations. Once the municipalities were identified, convenience sampling was employed to reach households. Eligible participants were identified from local municipality lists, and data collection ensued, ensuring the minimum required sample size was reached.

We refute the implication of selection bias and resulting bias. We have already stated our sampling strategy in which measures to achieve representativeness by geographical region were put in place; and as stated, geographical location is a proxy for known disparities in socio-economic and educational capital in Nepal; and all such regions were included. In following this strategy our sample should be at least as representative as many, if not most, population surveys. Further, we utilised random sample at the municipality level, as stated in the manuscript; another improvement on the methods of most population surveys which often allow participants to self-select with no elements of random sampling or stratification at all. We respectfully disagree with the reviewer’s comment, as we believe it is not accurate and the implication is not supported by the evidence presented.

The authors emphasize the need for early diagnosis and tailored interventions, particularly for older adults in higher altitude areas. However, the recommendations for public health policy are somewhat generic and lack specific, actionable steps.

Authors: Response to this comment has been addressed within the editor’s comments.

I am not sure I understand author’s justification of why prevalence is higher in women compared to men in Nepal.

Authors: Additional justification for the higher prevalence in women compared to men in Nepal is provided.

While the paper makes a valuable contribution to understanding dementia risk in Nepal, it requires more rigorous validation of its methods and a deeper exploration of confounding factors to strengthen its conclusions and recommendations.

Authors: We reiterate: The reviewer will be aware that all prevalence studies potential

---

## [Editor Report · Decision Letter 1]

7 Jul 2025

Risk of Dementia in Nepal: A Cross-Sectional Survey in Mountainous, Hilly, and Lowland Regions

PONE-D-24-59676R1

Dear Dr. Simkhada,

We’re pleased to inform you that your manuscript has been judged scientifically suitable for publication and will be formally accepted for publication once it meets all outstanding technical requirements.

Kind regards,

Saruna Ghimire

Academic Editor

PLOS ONE
---

## [Editor Report · Acceptance letter]

PONE-D-24-59676R1

PLOS ONE

Dear Dr. Simkhada,

I'm pleased to inform you that your manuscript has been deemed suitable for publication in PLOS ONE. Congratulations! Your manuscript is now being handed over to our production team.

Kind regards,

on behalf of

Dr. Saruna Ghimire

Academic Editor

PLOS ONE